# Pre-Hospital Management of Patients with COVID-19 and the Impact on Hospitalization

**DOI:** 10.3390/medicina59081440

**Published:** 2023-08-09

**Authors:** Floann Grannec, Line Meddeb, Herve Tissot-Dupont, Stephanie Gentile, Philippe Brouqui

**Affiliations:** 1IRD, MEPHI, IHU-Méditerranée Infection, Aix Marseille Université, 13005 Marseille, France; grannec.f@gmail.com; 2AP-HM, IHU-Méditerranée Infection, 13005 Marseille, France; line.meddeb@ap-hm.fr (L.M.); herve.tissotdupont@icloud.com (H.T.-D.); stephaniemarie.gentile@ap-hm.fr (S.G.); 3EA 3279 “Santé Publique, Maladies Chroniques et Qualité de Vie”, Aix Marseille University, 13005 Marseille, France

**Keywords:** pre-hospital managment, COVID-19, hospitalization, risk factor, time to care

## Abstract

*Background and Objectives:* During the COVID-19 pandemic, patient care was mainly organized around the hospital. Pre-hospital care has, to our knowledge, never been evaluated. We aimed to study the impact of pre-hospital pathways on hospitalization during the last part of the pandemic. *Materials and Methods:* This was a monocentric, retrospective analysis of prospectively collected medical records. Data from patients admitted to our institute between 1 February and 7 March 2022 were analyzed. The primary outcomes were defined as the number of hospitalizations, resuscitations, and deaths at the time of interview and in the subsequent 30 days. The main explanatory variables were times from onset of symptoms to care, age, gender, News2 score, comorbidities, and pre-hospital pathways and their duration. *Results:* Three pre-hospital pathways were identified: a pathway in which the patient consults a general practitioner for a test (PHP1); a pathway in which the patient consults for care (PHP2); and no pre-hospital pathway and direct admission to hospital (PHP3). Factors independently associated with outcome (hospitalization) were being male (OR 95% CI; 2.21 (1.01–4.84), *p* = 0,04), News2 score (OR 95% CI; 2.04 (1.65–2.51), *p* < 0.001), obesity (OR 95% CI; 3.45 (1.48–8.09), *p* = 0.005), D-dimers > 0.5 µg/mL (OR 95% CI; 3.45 (1.47–8.12), *p* = 0.005), and prolonged time from symptoms to hospital care (PHP duration) (OR 95% CI; 1.07 (1.01–1.14), *p* = 0.03). All things being equal, patients with a “PHP2” pre-hospital pathway had a higher probability of hospitalization compared to those with a “PHP3” pre-hospital pathway (OR 95% CI; 4.31 (1.48–12.55), *p* = 0.007). *Conclusions:* Along with recognized risk factors such as gender, News 2 score, and obesity, the patient’s pre-hospital pathway is an important risk factor associated with hospitalization.

## 1. Introduction

Since 2019, a new coronavirus infecting human has been actively circulating around the world. SARS-CoV-2, which is responsible for this pandemic, mainly causes upper and/or lower respiratory tract infections. A progression of these symptoms to more severe forms such as acute distress respiratory syndrome (ADRS) and eventually death is possible and is more likely to appear in patients with risk factors and/or comorbidities including obesity, diabetes, age, hypertension, and cancer [1]. “Mild” forms of this disease, mainly occurring in young patients, represent most cases, and their symptoms disappear after 5 to 14 days. However, silent or asymptomatic hypoxemia, a clinical presentation specifically reported in COVID-19, is present in one-third of patients upon hospital admission and is associated with a very poor outcome [2]. The lowest recorded pre-hospital oxygen saturation level (O2Sat) predicts death vs. discharge such that with every 1% reduction in the lowest O2Sat levels, there is a 13% increase in the odds of an in-hospital death [3]. The initial home O2Sat level predicts subsequent hospitalization [4]. Similarly, pulmonary embolism (PE) has been reported to have a 24% cumulative incidence in patients with COVID-19 pneumonia [5], and abnormal elevated D-dimer levels are associated with thrombosis, poor prognosis. and/or increased mortality [6]. D-dimer levels of >2.590 ng/mL were associated with a 17-fold increase in an adjusted risk of PE in hospitalized patients and was the strongest independent predictor of symptomatic venous thromboembolism (VTE) in patients with COVID-19 [7]. These features suggest that the administration of early VTE prophylaxis with either unfractionated heparin or low-molecular-weight heparin could be prescribed in outpatients with risk factors to prevent these thromboembolic phenomena [8]. Finally, there is increasing evidence that the use in the pre-hospital setting of the News2 score can predict critical illness and improve patient outcomes [9].

Consequently, early management at the pre-hospital stage of patients diagnosed with COVID-19, including evaluating clinical characteristics and comorbidities, screening for happy hypoxemia, NEWS2 score, oxygen saturation, D-dimer titration, by a health professional are all elements that can have an impact on the prognosis of the disease [10]. Pre-hospital treatment is likely to have an impact on hospitalizations, resuscitations, and deaths. In this study, we aimed to investigate the different pre-hospital pathways of COVID-19 patients treated at our institute [11] and to evaluate the impact of these pathways on outcomes in term of hospitalization, resuscitation, and death.

## 2. Material and Methods

The pre-hospital pathway (PHP) is defined as the set of stages the patient goes through from the onset of symptoms (OS) to their admission to hospital. These different stages generally involve the community health professionals in health facilities (pharmacies and health centers) as well as in medico-social structures when it comes to diagnostic or screening tests, various treatments, or monitoring (blood test and oxygen saturation). Some people prefer treatment at home involving consultations with general practitioners or specialists to receive treatment, diagnosis, and monitoring.

### 2.1. Ethical and Regulatory Aspects

The study was submitted for approval in advance to an Independent Ethical Committee (IEC), which gave a favorable opinion on 2 February 2022 (No. 2022-010). The IEC considered that this study does not fall under French bioethical law and complies with regulations on the processing of medical data and the Helsinki Declaration. The investigators’ declaration to comply with reference method MR 004 was filed prior to this study and was the subject of a declaration in the GDPR/APHM Register No. 2020-151.

### 2.2. Course of the Study and Data Collection

This is a retrospective study using medical data collected prospectively in 2022 by physicians during the medical treatment of patients presenting at the Institut Hospitalo-Universitaire (IHU) Méditerranée Infection for treatment after being diagnosed with COVID-19. Data recorded between 1 February and 7 March 2022 were selected and extracted from the hospital database for this study. Data from people who objected to their use were excluded.

The primary outcomes in terms of the variables of interest were defined as the number of hospitalizations, resuscitation, and deaths in the study population at the time of interview and in the subsequent 30 days. The explanatory variables were time from OS to treatment (calculated from the recorded dates of events), age, gender, News2 score, medical history (risk factors for COVID-19 disease: diabetes, elevated blood pressure, obesity, cancer, and previous episode of COVID-19), symptoms and treatments, as well as D-dimers levels. In order to complete the descriptive analysis, we asked the hospital information system to check whether any patients in the cohort had been hospitalized, transferred to intensive care, or died in the 30 days following this study.

### 2.3. Processing of Data and Variables Studied

For the purposes of the study, some data were recoded and, if necessary, created to identify the relevant explanatory variables. The missing data from the modalities of variables, such as the News2 score, were imputed so that the analyses were as relevant as possible.

### 2.4. Sample Size

An estimate of the number of individuals to include was calculated based on the prevalence of the primary outcome (hospitalization rate) in a COVID-19 cohort of >10,000 patients between wave 1 and wave 4 [11]. The average prevalence of hospitalization in this cohort was 2.7%, with a decrease of 30% in the last period. The sample size required for a descriptive study in a population of 30,000 patients previously followed in the IHU, with a main event frequency (hospitalization rate) of 2% and an acceptable margin of error of 5% for a 99.9% confidence interval, was 85 individuals.

### 2.5. Statistical Analysis

We first describe the population and the main PHPs with their respective duration between OS and presentation at hospital. In the second step, we evaluate the risk factors associated with the outcome using univariate logistic regression. We then use multivariate binary logistical regression to estimate the association between the risk of hospitalizations and these care pathways among the other risk factors recognized in the literature (age, obesity, comorbidities, treatments, etc.). Multivariate analysis was modelled using a step-by-step ascending binary logistic regression (Wald) and is confirmed by step-by-step descending binary logistic regression (Wald). Non-significant differences or associations were considered when the *p*-value was >0.05. Variables excluded from the multivariate model-analysis were those for which *p*-values were >0.20 in univariate analysis if data concerned only a sub-group of the total population or variable redundancy with others in the model. Univariate and multivariate analyses were performed using IBM SPSS Statistics.

## 3. Results

### 3.1. Data Quality and Sample Representativity

A total of 548 datasets collected from 548 consecutively interviewed patients between 1 February and 7 March 2022 were extracted from the database. Overall, 3 of these 548 patients refused the use of their data (0.55%), resulting in 545 being used for analysis. Among this cohort, 514 were seen as outpatients, and 31 were seen as hospitalized patients. A detailed analysis of missing data as well as a ratio of datasets available for analysis is presented in the flowchart Figure 1. 

### 3.2. Descriptive Analysis of Pre-Hospital Management

Of the 545 patients, 46 were hospitalized (11.8%), 1 died within the 30 days following hospitalization, and 1 was further admitted to the intensive care unit (ICU). Consequently, among the primary outcome variables, “hospitalization” only was retained for analysis. 

Three mains PHPs were characterized: PHP1, when patients consulted a health professional without having a positive diagnostic test (they were considered to be patients who were either symptomatic or asymptomatic (20.2%) and who came to the doctor to get a diagnostic test prescription); PHP2, when patients consulted a health professional with a positive test (they were considered to be patients who were either symptomatic or asymptomatic (4.7%) and who came to the doctor for treatment); and PHP3, when patients had seen no healthcare professionals in their PHP (Figure 2). 

PHP1 was undertaken by 129 patients, and 14 were hospitalized (10.9%). PHP2 was undertaken by 43 patients, and 12 were hospitalized (27.9%). PHP3 was undertaken by 373 patients, and 20 were hospitalized (5.4%). After checking the hospital information system, none of the 499 patients who received treatment at the outpatient department of the hospital were hospitalized with 30 days of the study period. The hospitalization rate was significantly higher in PHP2 than in PHP3 or PHP1 and higher in PHP1 than in PHP3. 

The time from OS to any consultation was significantly shorter for patients in PHP1 (0.41 +/− 4.8 days) compared to PHP2 (4.79 +/− 4.42 days; *p* < 0.001). Moreover, the time from the OS to hospital admission (PHP duration) was significantly longer in PHP2 (9.23 +/− 5.39 days) compared to PHP1 (6.24 +/− 4.82 days; *p* < 0.001) and to PHP3 (4.62 +/− 4.12; *p* < 0.001) (see Table 1, Figure 2). Indeed, patients who had seen a health professional (PHP2) arrived later at the hospital. Health monitoring by the GP, including monitoring of oxygen saturation levels (with a prescription for a pulse oximeter) and/or a blood test, was reported in only 21.9% of symptomatic patients. Although 77.7% to 85.7% of symptomatic patients received at least one medication from their GP, 73.1% were also treated while asymptomatic. Treatment by GPs mainly included azithromycin (39.5%), paracetamol (37.5%), zinc (32.6%), vitamins (20.9%), other antibiotics (19.2%), and corticosteroids in 17.4% of cases, while the treatment provided at the hospital included mainly hydroxychloroquine (62%), azithromycin (80.2%), zinc (82.6%), and anticoagulation (59.6%) (Table 2).

### 3.3. Univariate Analysis

In univariate analysis (Table 3), the probability of hospitalization was significantly higher among men (OR 95% CI; 1.95 (1.05–3.63), *p* = 0.023), older patients (OR 95% CI; 1.06 (1.03–1.08), *p* < 0.001), and those who presented with an elevated News2 score (OR 95% CI; 2.09 (1.73–2.52)), *p* < 0.001). Similarly, the probability of hospitalization was higher among obese patients (OR 95% CI; 2.17 (1.15–4.10), *p* = 0.015) and those with D-dimers levels > 0.5 µg/mL (OR 95% CI; 6.15 (2.9–12.7), *p* < 0.001). A longer time between OS and hospital admission (PHP duration) was associated with a higher probability of hospitalization (OR 95% CI; 1.12 (1.7–1.18), *p* = 0.03) as well as a longer time between the onset of symptoms to presentation to healthcare professional (OR 95% CI; 1.07 (1.01–1.13), *p* = 0.02). More than three days between OS and consultation with a GP was also associated with a higher probability of hospitalization (OR 95% CI; 5.63 (2.93–13.76), *p* = 0.02).

Surprisingly, monitoring (O2Sat/blood pressure/blood analysis, etc.) by the GP during PHP2 was associated with a higher probability of hospitalization (OR 95% CI; 5.33 (2.51–11.30), *p* < 0.001) as well as the prescription of any COVID treatment by the GP (OR 95% CI; 2.32 (1.23–4.33), *p* = 0.007). Among the different pre-hospital pathways, PHP2 and PHP1 were more frequently associated with a higher probability of hospitalization than PHP3 (OR 95% CI; 2.15 (1.05–1.39), *p* = 0.036) and (OR 95% CI; 6.83 (3.06–15.27), *p* < 0.001), respectively. 

### 3.4. Multivariate Analysis

Factors independently associated with the outcome (hospitalization) were being male (OR 95% CI; 2.21 (1.01–4.84), *p* = 0.04), News2 score (OR 95% CI; 2.04 (1.65–2.51), *p* < 0.001), obesity (OR 95% CI; 3.45 (1.48–8.09), *p* = 0.005), D-dimer levels > 0.5 µg/mL (OR 95% CI; 3.45 (1.47–8.12), *p* = 0.005), and prolonged time from symptoms to hospital care (PHP duration) (OR 95% CI; 1.07 (1.01–1.14), *p* = 0.03). All things being equal, patients with a PHP2 had a higher probability of hospitalization compared to the PHP3 pre-hospital pathway (OR 95% CI; 4.31 (1.48–12.55), *p* = 0.007).

## 4. Discussion

### Strengths and Weaknesses

To our knowledge, this study is the first to identify PHPs in COVID-19 patients. This study is monocentric, and consequently, the results presented below are applicable only to our sample. A multicenter study would have made it possible to draw general conclusions as to the observations made on the general population. However, we would have lost precision and would have taken the risk that no effect would be visible, in connection with Simpson’s paradox [12]. Although this is a retrospective study, memory bias is avoided since the data were collected prospectively by doctors using a standardized hospital electronic medical record system at the time of patient treatment, which in addition led to very few missing data. The results are highly significant, as evidenced by the univariate and multivariate analyses, which were obtained from detailed models and corrected with adjusted odds ratios. We originally aimed to study the impact of PHPs on hospitalizations, admissions to ICU, and deaths. This was impossible since, fortunately, only two patients met the last two criteria. Indeed, this study comes a little late in the unfolding of the pandemic. This study was conducted exclusively on the last wave, when the SARS-CoV2 Omicron variant was dominant. It is likely that access to pre-hospital care has changed during the different epidemic waves since 2020, as has the severity of the disease in relation to the viral genotype [13]. The impact of vaccination on at-risk patients may also have altered prognosis and, in particular, hospitalization. 

Our study reports that for patients with suspected or confirmed COVID, the shorter the PHP, the better the outcome. This has also been reported for other diseases, including HIV, sepsis, and cancer [14,15,16]. In COVID-19, delaying admission to hospital after the onset of the illness is associated with prolonged SARS-COV-2 shedding [17], and the time from OS to admission (PHP duration) is significantly associated with death [18]. Having a test prescribed by a GP (PHP1) as well as having treatment provide by a GP (PHP 2) is a risk factor for hospitalization. This paradoxical feature may be explained by the prolonged duration of the PHP or the fact that the GPs screen and refer to hospital only the most severe patients. The few deaths and resuscitations in this cohort did not enable us to answer the second question. Of the 499 patients who presented directly at the outpatient department of the hospital and returned home after being investigated and treated, none were subsequently hospitalized in the 30-day follow-up period. This ratio of hospitalization suggests that standardized treatment might be beneficial to the outcome. Pre-hospital monitoring was reported as lowering hospitalization (2.4 vs. 3.9%), ICU admission (0.3 vs. 1.1%), and 90 day mortality (0.2 vs. 0.6%) [19]. In another study assessing oximetry surveillance at home, the mortality after hospital admission was higher in those not monitored [20]. At the Mayo Clinic, the home monitoring system for high-risk patients showed comparable efficacy with a reduced hospitalization ratio (13.7 vs. 18%) and lower mortality (0.5 vs. 1.7%) [21]. A recent review is available on remote monitoring of COVID-19 patients [22]. Because the late management of patients with asymptomatic hypoxemia led to a very poor outcome [2], meticulous screening using home oximetry with virtual monitoring could prevent hospitalization, resuscitation, and/or death [3,23]. Some authors suggest using risk prediction scores including a simple interview or clinical examination features to guide hospitalization, such as, for example, the SOARS [24], the News2, and the RECAP [25] early-warning scores that can be used by GPs in the pre-hospital management of patients. The measurement of D-dimer levels in the early course of the disease showed that D-dimer levels of up to 1 µg/mL were associated with death and that preventive anticoagulation should improve outcomes [26,27]. Pre-hospital management of COVID-19 is likely to have changed during the epidemic with telemedicine and remote surveillance. In 2020, a prospective cohort study reported that, over 28 days of outpatient monitoring, 10.9% of the patients presented directly to the emergency department, and 7.6% required hospitalization [4].The role of pre-hospital management would be to avoid hospitalization, thus limiting hospital overcrowding and its consequences on overall care. Consequently, reducing the time between OS and primary care, the early assessment of disease prognosis based on scores, and the monitoring of prognosis-associated variables such as oxygen saturation and D-dimer levels by general practitioners could help guide hospitalization, relieving the hospital system while improving patient outcomes (Figure 3). Such pre-hospital care through specific outpatient care, ad hoc home visits such as “Coronataxi” [28], a screening and primary care structure, or specialized medical centers for general practitioners should be discussed in preparation for a new pandemic.

Implications for policy and practice:The role of pre-hospital management must be to avoid hospitalization, thus limiting hospital overcrowding and its consequences on overall care;An organized PHP, which reduces the time between OS and primary care; early assessment of disease prognosis based on scores; and community monitoring of prognosis-associated variables such as oxygen saturation and D-dimer levels by general practitioners can help guide hospitalization and relieve the hospital system while improving patient outcomes;Such pre-hospital care through specific outpatient treatment, ad hoc home visits, a screening and primary care structure, or specialized medical centers for general practitioners should be discussed in preparation for a new pandemic.

## Figures and Tables

**Figure 1 medicina-59-01440-f001:**
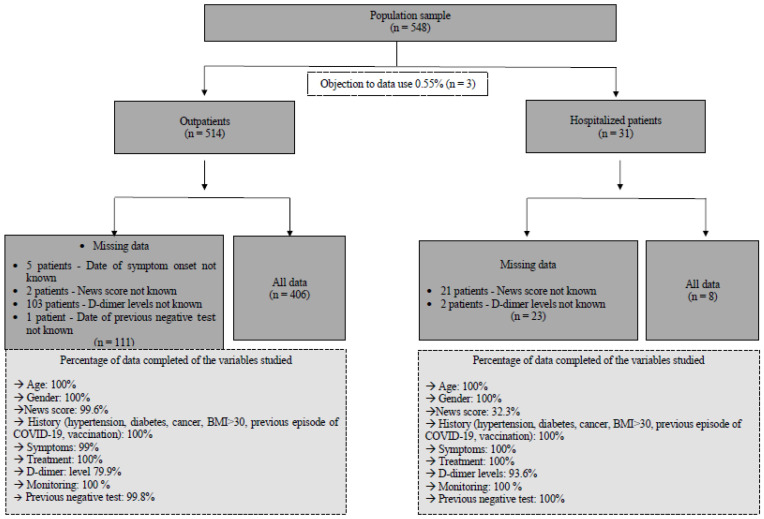
Flowchart describing the study population and indicating the reliability of the data used for descriptive, univariate, and multivariate analyses.

**Figure 2 medicina-59-01440-f002:**
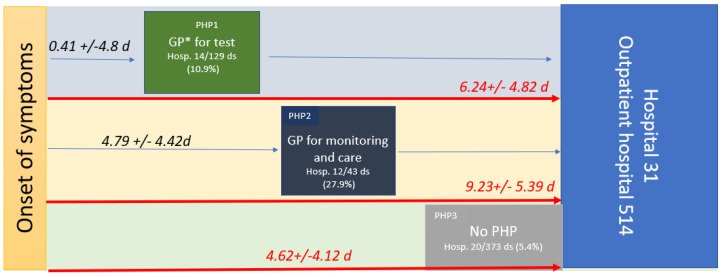
PHP, time from OS to care in symptomatic patients in days (median (range)), and hospitalization rates within the three pre-hospital pathways (PHP). PHP1 is green, PHP2 is dark blue, and PHP3 is grey. The number of datasets (ds) available for the study (545); * GP, general practitioner.

**Figure 3 medicina-59-01440-f003:**
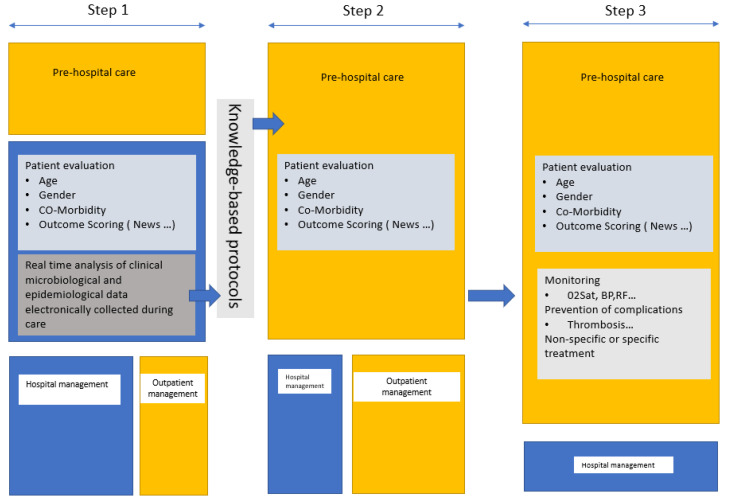
Proposed management in a future epidemic. The first step is mainly carried out in the reference hospitals (in blue) for clinical and epidemiological evaluation of the patient by specialists and the development and evaluation of microbiological diagnostic tests. Real-time data collection and automated analyses to identify risk factors for poor outcomes should guide clinical management. Knowledge-based protocols should be written for dissemination. The second step is pre-hospital triage (in yellow) and management by GPs based on the protocols issued by the reference centers. In step 3, in addition to pre-hospital triage, GPs can implement monitoring and prophylaxis and, where necessary, administer treatment.

**Table 1 medicina-59-01440-t001:** PHP duration.

	PHP3 No. = 362	PHP2 No. = 43	PHP1 No. = 129	Total	*p*-Value
Time between OS and any Consultation	4..62 +/− 4.12 (0–37)	4.79 +/− 4.42 (−1–20)	0.41 +/− 4.8 (−26–11)	3.64 +/− 4.6 (−26–37)	<0.001
	**	0.801	<0.001		
Time between OS and hospital care *	4.62 +/− 4.12 (0–37)	9.23 +/− 5.39 (0–24)	6.24 +/− 4.82 (0–27)	5.40 +/− 4.57 (0–37)	<0.001
	**	<0.001	<0.001		

* PHP duration; ** category of variable considered as reference in comparison with other categories.

**Table 2 medicina-59-01440-t002:** COVID-19 treatment administered by GPs and by the outpatient department of the hospital during the pre-hospital period.

	GP Prescriptions (172)	Out-Hospital Department Prescriptions (545)	*p*-Value *
	(Yes/no) %	(yes/no) %	
Paracetamol	(65/107) 37.8%	(0/545) 0.0%	<0.001
Azithromycin	(68/104) 39.5%	(437/108) 80.2%	<0.001
Other antibiotics	(33/139) 19.2%	(18/527) 3.3%	<0.001
Zinc	(56/116) 32.6%	(450/95) 82.6%	<0.001
Ivermectin	(11/161) 6.4%	(180/365) 33.0%	<0.001
Corticosteroids	(30/142) 17.4%	(13/532) 2.4%	<0.001
Hydroxychloroquine	(3/169) 1.7%	(338/207) 62.0%	<0.001
Anticoagulant	(10/162) 5.8%	(325/220) 59.6%	<0.001
Vitamin	(36/136) 20.9%	(0/545) 0.0%	<0.001

GP, general practitioner; * ANOVA.

**Table 3 medicina-59-01440-t003:** Factors associated with hospitalization among 545 consecutive patients interviewed at the hospital between 1 February and 7 March 2022.

Suspected Risk Factors	Outcome		Univariate Analysis		Multivariate Analysis	
	No Hospitalization, n = 499	Hospitalization, n = 46	*p*-Value	Odds Ratio and 95% CI	*p*-Value	Odds Ratio and 95% CI
*Gender **						
Male	221 (88.8)	28 (11.2)	0.023	1.95 (1.05–3.63)	*p* = 0.04	2.21 (1.01–4.84)
Female	278 (93.9)	18 (6.1)				
*Age **						
Med (range); Mean +/− SD	63 (18–92); 60.59 +/− 14.5	72 (33–89); 69.74 +/− 11.87	<0.001	1.06 (1.03–1.08)	NS	
Score_News *						
Med (range); Mean +/− SD	3 (0–9); 2.43 +/− 1.95	6 (1–11); 5.61 +/− 2.05	<0.001	2.09 (1.73–2.52)	<0.001	2.04 (1.65–2.51)
*Obesity **						
Obesity	106 (86.2)	17 (13.8)	0.015	2.17 (1.15–4.10)	0.005	3.45 (1.46–8.09)
No obesity	393 (93.1)	29 (6.9)				
*Comorbidity*						
Comorbidity	380 (91.1)	37 (8.9)	0.326	1.28 (0.60–2.75)	NA	
No comorbidity	119 (93)	9 (7)				
*D-dimer levels*						
**D-dimer > 0.5**	183 (83.6)	36 (16.4)	<0.001	6.15 (2.9–12.7)	*p* = 0.005	3.45 (1.47–8.12)
**D-dimer < 0.5**	313 (96.9)	10 (3.1)				
*Time between OS and admission to hospital care? (PHP duration) **						
Med (range); Mean +/− SD	4 (0–37); 5.10 +/− 4.28	7 (0–25); 8.72 +/− 6.13	<0.001	1.12 (1.7–1.18)	0.03	1.07 (1.01–1.14)
*Time between OS and any consultation*						
Med (range); Mean +/− SD	3 (–26–37); 3.5 +/− 4.5	4.5 (−5–20); 5.15 +/− 5.4	0.02	1.07 (1.01–1.13)	NA	
*Time between OS and GP consultation*						
Time > 3 days	22 (62.9)	13 (37.1)	0.02	5.63 (2.93–13.76)	NA	
Time < 3 days	124 (90.5)	13 (9.4)				
Monitoring by GP						
Monitoring by GP	31 (72.1)	12 (27.9)	<0.001	5.33 (2.51–11.30)	NA	
No monitoring by GP	468 (93.2)	34 (6.8)				
*COVID treatment by GP **						
COVID treatment by GP	116 (85.9)	19 (14.1)	0.007	2.32 (1.23–4.33)	NS	
No COVID treatment by GP	383 (93.4)	27 (6.6)				
*Pre-hospital pathways **			<0.001		0.02	
PHP1	115 (89.1)	14 (10.9)	0.036	2.15 (1.05–4.39)	0.10	2.13 (0.86–5.27)
PHP2	31 (72.1)	12 (27.9)	<0.001	6.83 (3.06–15.27)	0.007	4.3 (1.48–12.55)
PHP3	353 (94.6)	20 (5.4)	-	-	-	-

* Variables selected to be included in multivariate model analysis.

## Data Availability

Data are available on demand to the corresponding author.

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
