# Peer review of "Pre-Hospital Management of Patients with COVID-19 and the Impact on Hospitalization"

_medicina, 2023, doi:10.3390/medicina59081440_

Round 1

Reviewer 1 Report

Dear authors,

This subject still arouses the interest of professionals in the medical field. I congratulate you for the work you have done and for the chosen subject.

Back to article:

- the abstract meets the editorial requirements of the journal, is less than 300 words and is organized as a single paragraph containing: 1) Background and objectives, 2) Materials and methods, 3) Results and 4) Conclusion

- the introduction, material and method and the results are reproduced accordingly.

- I would insist on the discussion section. The current literature abounds in research papers and reviews on the therapy applied to the patient in the pre-hospital, but also if it requires hospitalization and the reasons for which the decision was made.

- also, 23 titles are insufficient for the references of a paper about the infection with the new coronavirus

Good luck!

Author Response

Reviewer 1

This subject still arouses the interest of professionals in the medical field. I congratulate you for the work you have done and for the chosen subject.

A1: thank you for congratulation.

The abstract meets the editorial requirements of the journal, is less than 300 words and is organized as a single paragraph containing: 1) Background and objectives, 2) Materials and methods, 3) Results and 4) Conclusion.

A2: We reorganize and modified the abstract as required. The number of words is now 277

Abstract:

Background and objectives: During the COVID-19 pandemic, patient care was mainly organized around the hospital. Pre-hospital care has, to our knowledge, never been evaluated. We aim to study the impact of pre-hospital pathways on hospitalization during the last part of the pandemic. Materials and methods: This was a monocentric retrospective analysis of prospectively collected medical records. Data from patients admitted to our institute between 1 February and 7 March 2022 were analyzed. The primary outcomes were defined as the number of hospitalizations, resuscitations, and deaths at the time of interview and in the subsequent 30 days. The main explanatory variables were times from onset of symptoms to care, age, gender, News2 score, comorbidities, and pre-hospital pathways and their duration. Results: Three pre-hospital pathways have been identified: a pathway in which the patient consults a general practitioner for a test (PHP1); a pathway in which the patient consulted for care (PHP2); and no pre-hospital pathway and direct admission to hospital (PHP3). Factors independently associated with outcome (hospitalization) were being male (OR 95% CI; 2.21 [1.01–4.84], p=0,04), News2 score (OR 95% CI; 2.04 [1.65–2.51], p<0.001), obesity (OR 95% CI; 3.45 [1.48–8.09], p=0.005), D-dimers > 0.5 µg/ml (OR 95% CI; 3.45 [1.47–8.12], p=0.005), prolonged time from symptoms to hospital care (PHP duration) (OR 95% CI; 1.07 [1.01–1.14], p=0.03). All things being equal, patients with a “PHP2” pre-hospital pathway had a higher probability of hospitalization compared to those with a “PHP3” pre-hospital pathway (OR 95% CI; 4.31 [1.48–12.55], p=0.007). Conclusions. Along with recognized risk factors such as gender, News 2 score, and obesity, the patient’s pre-hospital pathway is an important risk factor associated with hospitalization.

The introduction, material and method and the results are reproduced accordingly.

A3: Thank you.

I would insist on the discussion section. The current literature abounds in research papers and reviews on the therapy applied to the patient in the pre-hospital, but also if it requires hospitalization and the reasons for which the decision was made.

- also, 23 titles are insufficient for the references of a paper about the infection with the new coronavirus

A4: We thank the reviewer that help to complete the discussion accordingly and we add the needed reference. We found 3 papers and a review that report on the efficacy of prehospital management and hospitalisation ICU and death in covid that we add in the discussion.

"Pre-hospital monitoring was reported as lowering hospitalization (2.4 vs 3.9%) , ICU admission (0.3 vs 1.1%) and 90 day mortality ( 0.2 vs 0.6 %) (17). In another study assessing oximetry surveillance at home, the mortality after hospital admission was higher in those not monitored (18). At the Mayo clinic, the home monitoring system for high risk patients showed comparable efficacy  with a reduced hospitalization ratio ( 13.7 vs 18%)  and lower mortality ( 0.5 vs 1.7%) (19). A recent review is available on remote monitoring of Covid -19 patients (20)."

Reviewer 2 Report

The topic is of great interest. However, it is unclear how and why the cohort for the analysis was chosen. Also, the findings that patients with more severe disease (need for oxygen monitoring or antibiotic treatment) are more likely to be   hospitalized is not really a novel finding and not really surprising. The same is true for patients with milder symptoms, who did need to see a health care provider and less frequently were hospitalized. 

The chosen cohort also limits interpretation of other factors that might indicate other/new risk factors for more severe course.

The paper would benefit also from native English editing.

Author Response

Reviewer 2 :

The topic is of great interest.

A1: Thank you

However, it is unclear how and why the cohort for the analysis was chosen.

A2: During the last flu pandemic, the role of general practitioner and prehospital management was questioned. During the last wave of COVID-19 one question arise. Did we do our best to early manage the outbreak? and was our preparedness plan was appropriated? Consequently we wanted to have a look back to prehospital pathway of patient admitted to our institution. The hospitalization rate being high we needed only a few patients and so we were interested in the most recent patients, the medical questionnaires having expanded during the pandemic it was in these that we had the most information available.   

Also, the findings that patients with more severe disease (need for oxygen monitoring or antibiotic treatment) are more likely to be   hospitalized is not really a novel finding and not really surprising. The same is true for patients with milder symptoms, who did need to see a health care provider and less frequently were hospitalized. 

A3: In the multi variate analysis, the pathway is independently linked to hospitalization. This means that even if severe disease as measured by the News2 Score, gender, obesity and D-Dimers were also independently associated with hospitalization,  the prehospital pathway was a non-negligible factor associated to hospitalization with a risk up to 4 times for patient having a PHP2 rather than a PHP3. This is similar to other study which have also found that prehospital home monitoring lowered hospital admission, ICU and death. We add in the discussion the data from this study.

The chosen cohort also limits interpretation of other factors that might indicate other/new risk factors for more severe course.

A5: We agree. This is due to the retrospective nature of the study in which unfortunately contained lacking data.  

Comments on the Quality of English Language

The paper would benefit also from native English editing.

A6: The paper was edited in English by TraDonline ®

Round 2

Reviewer 1 Report

Dear authors,

thank you for revised after recommendations this article. I still believe that can be upgraded by new references. But I appreciate your effort to ameliorate this.

Good luck!